# Non-glycosylated IGF2 prohormones are more mitogenic than native IGF2

Pavlo Potalitsyn [1,2], Lucie Mrázková[1,3], Irena Selicharová[1], Michaela Tencerová [4], Michaela Ferenčáková[4], Martina Chrudinová[1], Tereza Turnovská[1], Andrzej Marek Brzozowski [5], Aleš Marek[1], Jakub Kaminský[1], Jiří Jiráček [1✉] & Lenka Žáková [1✉]

Insulin-like Growth Factor-2 (IGF2) is important for the regulation of human embryonic growth and development, and for adults' physiology. Incorrect processing of the IGF2 precursor, pro-IGF2(156), leads to the formation of two IGF2 proforms, big-IGF2(87) and big-IGF2(104). Unprocessed and mainly non-glycosylated IGF2 proforms are found at abnormally high levels in certain diseases, but their mode of action is still unclear. Here, we found that pro-IGF2(156) has the lowest ability to form its inactivating complexes with IGF-Binding Proteins and has higher proliferative properties in cells than IGF2 and other IGF prohormones. We also showed that big-IGF2(104) has a seven-fold higher binding affinity for the IGF2 receptor than IGF2, and that pro-IGF2(87) binds and activates specific receptors and stimulates cell growth similarly to the mature IGF2. The properties of these pro-IGF2 forms, especially of pro-IGF2(156) and big-IGF2(104), indicate them as hormones that may be associated with human diseases related to the accumulation of IGF-2 proforms in the circulation.

[1] Institute of Organic Chemistry and Biochemistry, Czech Academy of Sciences, Flemingovo nám. 2, 116 10, Prague 6, Czech Republic. [2] Department of Biochemistry, Faculty of Science, Charles University, 12800 Prague 2, Czech Republic. [3] Department of Cell Biology, Faculty of Science, Charles University, 12800 Prague 2, Czech Republic. [4] Institute of Physiology, Czech Academy of Sciences, Vídeňská 1083, Prague 4, Czech Republic. [5] York Structural Biology Laboratory, Department of Chemistry, University of York, Heslington, York YO10 5DD, UK. ✉email: jiracek@uochb.cas.cz; zakova@uochb.cas.cz

Although insulin-like growth factor 2 (IGF2) was characterized several decades ago, its physiological role is still largely unknown. IGF2 is expressed by most tissues in the body but plays a role especially in prenatal growth and development. Other tissues affected by IGF2 levels are the placenta, brain, skeletal muscle and bone[1–3]. Most of IGF2's biological actions are mediated by its binding and signaling through three structurally homologous receptors: IGF1 receptor (IGF1R), two isoforms of insulin receptors (IR-A and IR-B) and a structurally distinct mannose-6-phosphate/IGF2 receptor (M6P/IGF2R)[1,4]. Mature IGF2 is a 67-amino-acid protein that is formed by the post-translational removal of the C-terminal E-domain from the precursor, the 156-amino-acid-long pro-IGF2(156)[5] (Fig. 1). Under normal conditions, pro-IGF2(156) is specifically endo-cleaved by proprotein convertases to form the mature IGF2. Endoproteolysis occurs concomitant to—or slightly after—the glycosylation maturation of pro-IGF2(156)[5], resulting in the secretion of two O-glycosylated molecules, called big-IGF2: big-IGF2(87) and big-IGF2(104). These two proteins, as well as pro-IGF2(156), are found in circulation along with native fully mature IGF2[6]. These longer, unprocessed forms represent 10–20% of the total IGF2 in normal human serum[7,8]. However, a much bigger proportion of incompletely processed IGF2 proforms (usually >60%) has been reported in certain pathophysiological conditions, such as non-islet-cell tumor hypoglycemia (NICTH) or hepatitis C-associated osteosclerosis (HCAO)[9–18]. Importantly, it has been demonstrated that these proforms, occurring in large, slowly growing solitary fibrous tumors and hepatoma carcinomas, are predominantly present there in non-glycosylated forms. This contrasts with the normal tissues, where the glycosylated forms of these IGF2 proforms prevail[13,18–22].

Normally, nearly 95% of IGF2 (and IGF1 as well) is bound to the IGF-Binding Protein-3 (IGFBP3), the most abundant IGFBP in serum, to form a 50 kDa binary complex[1]. This complex can subsequently bind leucine-rich glycoprotein acid-labile subunit (ALS) protein to form a 150 kDa ternary complex[23,24], responsible for the maintenance of ~70–80% of both IGFs in human serum and the circulatory regulation of the bioavailability of these hormones. The formation of binary and ternary complexes fundamentally determines the bioavailability of circulating IGF2 levels; the 150 kDa ternary complex has a long half-life (~16 h) and limited bioactivity in the tissues, due to the size that limits its passage through the capillaries[18,25], while the 50 kDa binary complex has a shorter half-life (20–30 min), but is small enough to cross capillary walls, gaining access to most tissues[17,26]. It has been reported that pro- and big-IGF2s have altered abilities to form these complexes, which may increase the amount of available free hormones in the circulation[17,18]. Moreover, pro- and big-IGF2 may suppress levels of growth hormone (GH) secreted from pituitary gland and insulin from pancreas[22], which subsequently decreases the synthesis and secretion of ALS, IGF1, and IGFBP3[14,27,28], promoting even higher bioavailability of the IGF2 proforms.

More than 25 years ago, it was hypothesized that non-glycosylated forms of IGF2 might have stronger proliferative properties than their glycosylated counterparts[13,20,21,29], but the molecular mechanisms by which these effects might be mediated are not known. Moreover, the previous studies[7,10,16–19] that demonstrated an abnormal occurrence of pro- and big-IGF2s have mostly been unable to distinguish between the exact proforms of IGF2. This was, and still is, challenging due to a varying degree of glycosylation affecting the molecular weight of IGF2 proforms[30]. In this study, we produced all three IGF2 proforms in their non-glycosylated forms that are suspected of having stronger proliferative properties than their glycosylated counterparts. We systematically compared their binding and activation abilities for the IGF and insulin receptors, intracellular signaling proteins, their abilities to form binary and ternary complexes with IGFBPs/ALS, proliferative capacities and effects on calcium transport in the specific permanent and primary cells. To make the picture more complete, we compared some of the binding and proliferation properties of pro-IGF2(156) with its commercially available glycosylated form. Since IGF2 is also critical for skeletal growth and maintenance[31,32], and big-IGF2s have been reported as important factors in HCAO, we also focused this research on the effect of IGF2 on different bone cell types and on the effect on calcium ion transport. This comprehensive study and the results have clarified and improved the current knowledge of the biological importance and pathogenicity of the individual IGF2-derived proforms of this hormone.

## Results

### Optimized production of pro-IGF2 and big-IGFs eliminated their misfolding.
Native IGF2, pro-IGF2, and both big-IGF2s were successfully cloned into the pRSFDuet vector and expressed

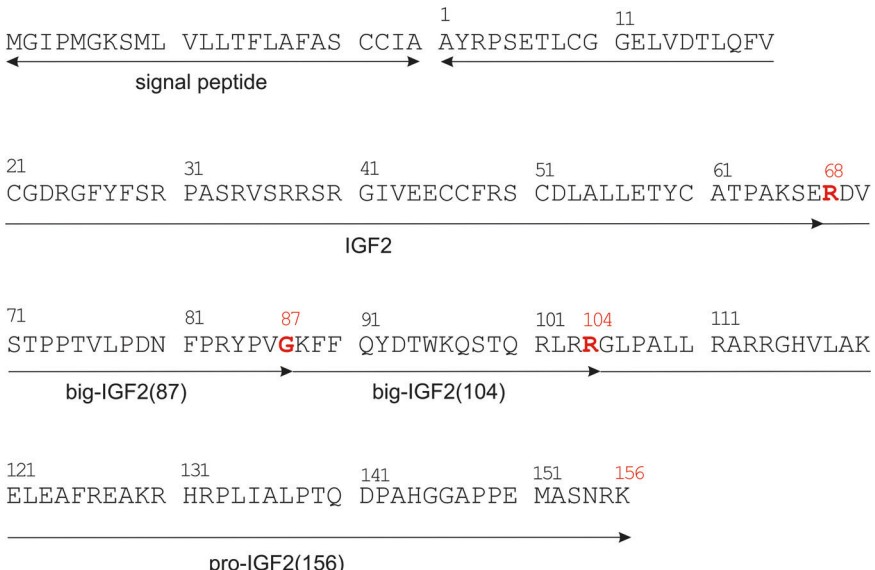

**Fig. 1 Primary sequence of pre-pro-IGF2.** Sites where pro-IGF2 is cleaved into big-IGF2(87) and big-IGF2(104) are marked in red.

to inclusion bodies, as described by Potalitsyn et al.[33]. Native IGF2 was purified from inclusion bodies, as previously described by Hexnerova et al.[34], yielding 0.25–0.6 mg of IGF2 protein per 1 L of cultivation media. However, despite the same methodology, the yields of pro-IGF2 and big-IGF2s were approximately only about one-half of that of the IGF2, ranging from 0.1 to 0.4 mg per 1 L of media, mainly due to misfolding of the prohormones. Thus, the methodology was modified by the adaptation of pH values of the buffers to the pI values of the prohormones' constructs, and, subsequently, any misfolded pro- or big-IGF2 forms were not observed. The purity of the analogs was >95% (analytical RP-HPLC) and identities were confirmed by high-resolution ESI-MS or MALDI (Supplementary Figs. S1, S2).

**Circular dichroic spectra of pro- and big-IGF2s indicate similar 3D structures**. Although several IGF2 structures are available (e.g. PDB IDs 2L29 and 5L3L), there is no structure (X-ray, NMR or cryo-EM) of big-IGF2(87), big-IGF2(104) or pro-IGF2(156). Therefore, to gain any insight into the structure of our proteins, we assessed their secondary structure by circular dichroism (CD) spectra in the far-UV region (Fig. 2) (A detailed description of the determination of secondary structures is in Supplementary Tables S1–S2, Supplementary Figs. S3–S5, and Supplementary Note 1). Spectra and their analysis indicate that the secondary structures between Ala1 and Glu67 in the core parts of big-IGF2s and pro-IGF2 are largely similar to IGF2, whereas their remaining parts (extra E-domains) are rather unstructured.

**Big-IGF2(104) activates IR-A more efficiently than mature IGF2**. Big-IGF2(104) and big-IGF2(87) bind both isoforms of the insulin receptor (IR-A, IR-B) similarly to native IGF2, while pro-IGF2(156) had a significantly lower affinity for IR-A; a similar, but not so marked trend was found for binding of these IGF2 proforms to IR-B (Fig. 3a–d, Supplementary Figs. S6, S12 and Table 1 and Supplementary Table S3). To complement these assays, we performed a binding experiment with a commercially available glycosylated (gly) pro-IGF2(156) to IR-A. We found that its binding affinity is comparable to our non-glycosylated form (Table 1, Supplementary Tables S3 and S11A). The ability of the ligands (at 10 nM concentrations) to induce IR-A and IR-B autophosphorylation and phosphorylation of Akt paralleled their binding affinities, with the exception of big-IGF2(104) that was able to induce significantly higher phosphorylation of both receptors than IGF2 (Fig. 3b, d). Big-IGF2(104) also has the highest ability to activate Akt in IR-A cells of all IGF2 hormone variants, but the phosphorylation of Erk 1/2 induced by big-IGF2s and pro-IGF2 was similar to the effects of IGF2 in both the IR-A and IR-B cells. This suggests that big-IGF2(104) may have

the most important role in insulin receptor binding and activation in vivo of all IGF2 forms.

**Binding affinities of pro- and big-IGF2s to IGF1R decrease with increasing E-domain length**. The binding and activation of the IGF2 proforms were also investigated in the context of the IGF1 receptor. Here, the binding affinities of pro-IGF2(156) and both big-IGF2s to IGF1R (Fig. 3e and Supplementary Fig. S6) are markedly affected and related to their molecular weights, hence possible structural variations, i.e. pro-IGF2(156) has the lowest IGF1R affinity of the three IGF2 proforms, while big-IGF2(87) has a similar binding affinity to the native IGF2. Glycosylated pro-IGF2(156), for which we also measured its binding affinity to IGF-1R, has similar affinity as its non-glycosylated analogue, pro-IGF2(156) (Table 1, Supplementary Fig. S11B). However, big-IGF2(87) and big-IGF2(104) show similar stimulation of IGF1R autophosphorylation to IGF2, while the activation of this receptor by pro-IGF2(156) is apparently lower (Fig. 3e, Supplementary Fig. S12). However, it seems that such a decrease is not statistically significant. In the downstream IGF1R signaling, the phosphorylation of Akt decreases with the increase of the chain length of the ligands (following a similar trend observed for the binding affinities), but this decrease is statistically significant only for pro-IGF2(156). Phosphorylation of Erk 1/2 was induced similarly by IGF2, big-IGF2(87) and big-IGF2(104), while it was lower for pro-IGF2(156), although not statistically significant (Fig. 3f).

**Big-IGF2(104) binds M6P/IGF2R markedly more strongly than mature IGF2 and other IGF2 proforms**. Although the IGF2 proforms were tested for binding to immobilized D11:M6P/IGF2R, the main binding domain of IGF2 on M6P/IGF2R[33], we also developed and established a methodology to measure the binding affinity for the native M6P/IGF2R in the R- cells (Fig. 4a, b, Supplementary Figs. S6, S10, and S11C and Table 1 and Supplementary Table S3), which are 3T3-like fibroblasts derived from mouse embryos with a targeted disruption of Igf1r gene. R- cells express only negligible amounts of IGF1R and IR-A, but they have an abundant population of M6P/IGF2R. Therefore, these cells can serve as a model to study intracellular signaling induced by the interaction of our IGF2 proforms with the M6P/IGF2R without interference of IR and IGF-1R signaling. To confirm this, we performed a saturation binding experiment to determine the dissociation constant ($K_d$) of [125I]-monoiodotyrosyl-Tyr2-IGF2 for the native M6P/IGF2R in R- cells. To check whether these cells also contain amounts of IR or IGF1R that may interfere with M6P/IGF2R hormones' binding, the same experiment was also carried out with [125I]-monoiodotyrosyl-TyrA14-HI and [125I]-iodotyrosyl-IGF1, respectively (Supplementary Fig. S7). This gave a $K_d$ value of 8.2 nM for [125I]-IGF2 binding to M6P/IGF2R and

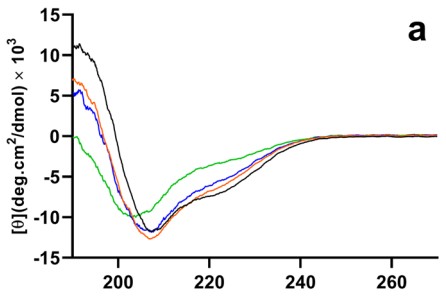

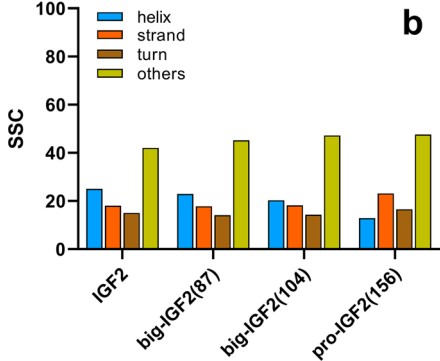

**Fig. 2 Experimental CD spectra of IGF2 derivatives.** CD spectra of IGF2 (black), big-IGF2(87) (red), big-IGF2(104) (blue), and pro-IGF2(156) (green) recorded in 0.1% $CH_3COOH$ (**a**). Secondary structure content (SSC, in %) estimated using BeStSel according to CD spectra (**b**).

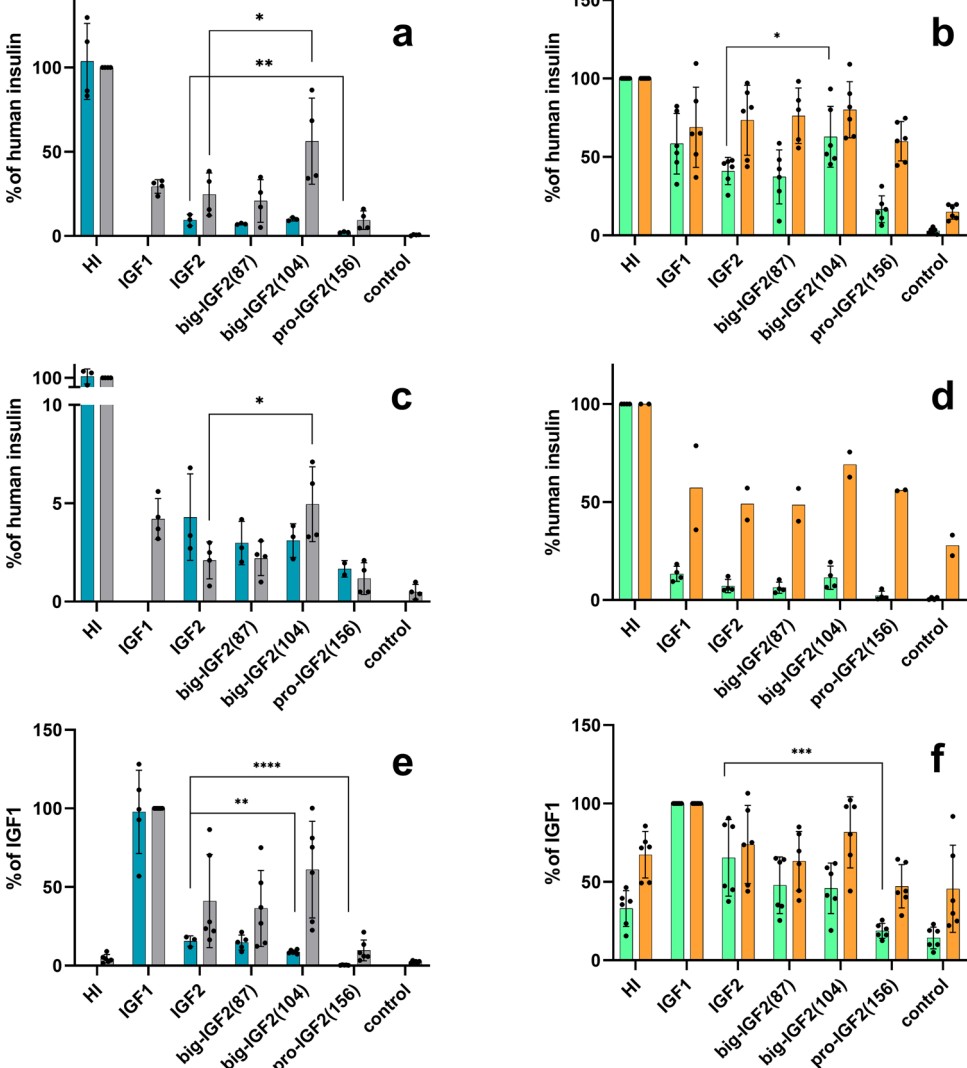

**Fig. 3 Binding affinities and receptor activation abilities of IGF2 derivatives for IR-A, IR-B and IGF-1R.** Binding affinity (cyan) and receptor autophosphorylation (gray) of human insulin (HI), IGF1, IGF2 and big-IGF2(87), big-IGF2(104), and pro-IGF2(156) to IR-A (**a**), IR-B (**c**) and IGF-1R (**e**); Phosphorylation of Akt (green) and Erk 1/2 (orange) on IR-A (**b**), IR-B (**d**) and IGF-1R (**f**) overexpressing cells. Data are presented as means ± S.D. relatively to the binding and stimulation induced by insulin (on IR-A and IR-B) or IGF1 (on IGF1R). Significant differences between marked values determined using Ordinary one-way ANOVA (Dunnett´s multiple comparisons test), *$p < 0.05$, **$p < 0.01$, ***$p < 0.001$, ****$p < 0.0001$.

**Table 1 Relative binding affinities (in %) of individual ligands for individual receptors and binding proteins.**

| Hormone | Receptor Relative binding affinity (%) | | | | | | |
|---|---|---|---|---|---|---|---|
| | IR-A | IR-B | IGF1R | M6P/IGF2R:D11 | M6P/IGF2R | IGBP3 (binary complex) | IGFBP3:ALS (ternary complex) |
| IGF2 | 8.5 ± 3.8 | 3.6 ± 1.8 | 15.8 ± 3.6 | 100 ± 30 | 100 ± 40 | 100 ± 71 | 100 ± 45 |
| IGF1 | 1.1 ± 0.4 | 0.11 ± 0.03 | 100 ± 47 | no binding | 4.0 ± 1.2 | 65 ± 35 | n.d. |
| HI | 100 ± 31 | 100 ± 37 | 0.08 ± 0.03 | n.d. | n.d. | n.d. | n.d. |
| big-IGF2(87) | 7.1 ± 1.6 | 2.7 ± 1.2 | 14.8 ± 7.1 | 73 ± 34 | 227 ± 119 | 70 ± 53 | 58 ± 40 |
| big-IGF2(104) | 9.8 ± 2.4 | 2.9 ± 1.2 | 9.7 ± 3.4 | 698 ± 295 | 887 ± 309 | 146 ± 73 | 17.0 ± 6.2 |
| pro-IGF2(156) | 2.2 ± 0.6 | 1.6 ± 0.6 | 0.35 ± 0.18 | 56 ± 42 | 20 ± 16 | 16.0 ± 11.3 | 9.1 ± 3.9 |
| gly pro-IGF2(156) | 2 | n.d. | 0.2 | n.d. | <1 | 18 | n.d. |

Relative binding affinities were calculated as ($K_d$ of the native hormone/$K_d$ of analog) × 100 (%). The experimental $K_d$ values are provided in ESI. The statistical evaluation of data is shown in Fig. 3. n.d. is not determined. gly means glycosylated. Data on gly pro-IGF2(156) were obtained from only one measurement due to the small amount of material available.

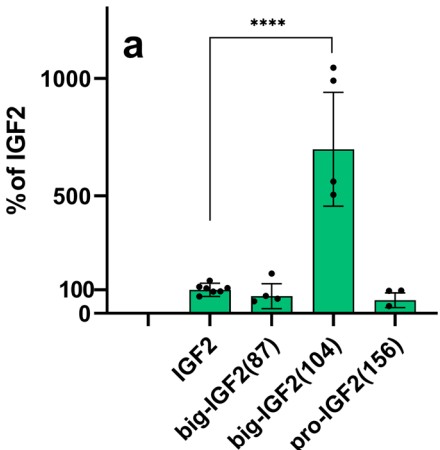
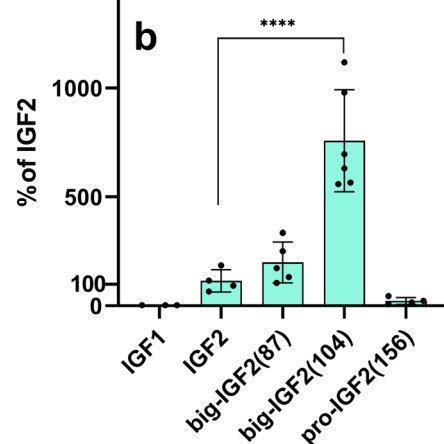

**Fig. 4 Relative binding affinities of IGF2 derivatives for M6P/IGF2R. a** Binding to immobilized domain 11 of M6P/IGF2R; **b** Binding to M6P/IGF2R in R-cells of IGF1, IGF2, big-IGF2(87), big-IGF2(104), and pro-IGF2(156). Data are presented as means ± S.D., relative to the binding affinity of IGF2. Significant differences between marked values determined using Ordinary one-way ANOVA (Dunnett´s multiple comparisons test), ****$p < 0.0001$.

$B_{max}$ value – proportional to the number of receptors – of about 1,378,000. In contrast, the numbers of binding sites for IGF1 ($B_{max} = 36,174$) and insulin ($B_{max} = 8145$) were significantly lower, not allowing the determination of the $K_d$ values for radio-labeled IGF1 and insulin.

The binding affinities of the hormones in these two systems, i.e. whole cell M6P/IGF2R receptor and immobilized D11 domain only, were fully comparable. IGF2 and big-IGF2(87) had similar affinities for both D11 and M6P/IGF2R, and pro-IGF2(156) showed slightly lower affinities than IGF2 in both binding systems. The binding of glycosylated pro-IGF2(156) to M6P/IGF2R is the weakest of all hormones tested and the protein was only able to displace $^{125}$I-IGF2 by 50%. Surprisingly, big-IGF2(104) had a more than sevenfold higher affinity for M6P/IGF2R than IGF2 and the other IGF2 proforms. This unexpected effect was also similar for immobilized D11 (Fig. 4, Supplementary Figs. S6, S10, S11, Table 1, and Supplementary Table S3).

Interestingly, the analysis of these data revealed a two-site binding mode of big-IGF2(87) and big-IGF2(104) in binding experiments to the whole M6P/IGF2R on R- cells, with some indication of a similar two-site binding of IGF2 as well; however, this phenomenon was not observed for pro-IGF2(156) (Supplementary Fig. S6D).

**Pro- and big-IGF2s can form binary and ternary complexes**. The abilities of IGF2 and its proforms to form binary complexes with IGFBP3 and ternary complexes with IGFBP3:ALS were also investigated. Firstly, saturation binding experiments of [$^{125}$I]-monoiodotyrosyl-Tyr2-IGF2 to immobilized IGFBP3 (binary complex) to determine this $K_d$ were carried out, followed by the evaluation of the $K_d$ value for [$^{125}$I]-monoiodotyrosyl-Tyr2-IGF2 binding to IGFBP3:ALS protein complex (a ternary complex, immobilized through His-tag of ALS) (Supplementary Figs. S8 and S9). The $K_d$ values of [$^{125}$I]-monoiodotyrosyl-Tyr2-IGF2 for the formation of binary and ternary complexes were 0.56 nM for and 5.9 nM, respectively. Subsequently, these values were used to calculate the affinities of the non-labeled (pro)hormones for binary and ternary complexes from competitive binding experiments. Here, the ability of big-IGF2(87) to bind IGFBP3 was apparently decreased, while it was increased for big-IGF2(104) in comparison to IGF2, but these differences were not statistically significant. In contrast, the ability of big-IGF2(156) to form a binary complex was significantly decreased, when compared to IGF2. Glycosylation appears to play a negligible role here, as we obtained a very similar result with glycosylated pro-

IGF2(156) (Fig. 5a, Supplementary Fig. S11D Table 1, and Supplementary Table S3). This indicated that the abilities of these (pro)hormones to form ternary complexes with IGFBP3:ALS decrease with the increase of the length of their E-chains (Fig. 5b and Table 1).

**Pro-IGF2(156) can stimulate phosphorylation of Erk 1/2 in R-cells and MC3T3-E1 preosteoblast more efficiently than other IGF-2 proforms**. The expression of IR-A, IR-B, IGF1R and M6P/IGF2R in the cell lines used in this study was investigated for signaling experiments and calcium release (R-, R+39, IR-A, IR-B, U2OS, and MC3T3-E1 cells). The M6P/IGF2R expression was comparable in all cell types (Supplementary Fig. S13), with it being relatively highest in the R- cells. As expected, there was no IGF1R in the R-cells and in the insulin receptor transfected IR-A and IR-B cells. The expression of IGF1R was very large in the R +39 cells, but only its very low levels were detected in the E1 cells, with a slightly higher level in the U2OS (about 4x more when the band density is normalized to the actin level). As expected, a high expression of IR was observed in the transfected cells IR-A and IR-B, with almost no signal for this receptor in the other cell lines. Only very overloaded gels indicated some IR expression in E1, U2OS and R- cells, and the expression in E1 cells was slightly higher than in U2OS. The R- cells (convenient model cells to study signaling through M6P/IGF2R) and murine MC3T3-E1 preosteoblast cells (with natural receptor populations and with the high abundance of M6P/IGF2R) were used, to investigate the effects of IGF2, big-IGF2s and pro-IGF2(156) on the activation of the intracellular Akt and Erk 1/2 proteins (Fig. 6).

Due to the absence of signaling domain in M6P/IGF2R, which is a predominant receptor for IGF2 in R- cells, low effects on both investigated pathways were expected. Indeed, phosphorylation of the Akt and Erk 1/2 induced by IGF2, big-IGF2(87) and big-IGF2(104) were modest. However, surprisingly, a strong phosphorylation of Erk 1/2 after stimulation with pro-IGF2(156) in R- cells was observed (Fig. 6a). A similar effect was seen in MC3T3-E1 preosteoblasts, where big-IGF2(156) induced significantly higher phosphorylation of Erk 1/2 than any other ligand. On the other hand, phosphorylation of the Akt by all prohormones and IGF2 in MC3T3-E1 cells was similar, not significantly different from the control (Fig. 6b). Thus, pro-IGF2(156) appears to have clearly significantly stronger effects on Erk 1/2 activation in preosteoblasts and IGF1R-deficient fibroblasts than other tested IGF-related proteins.

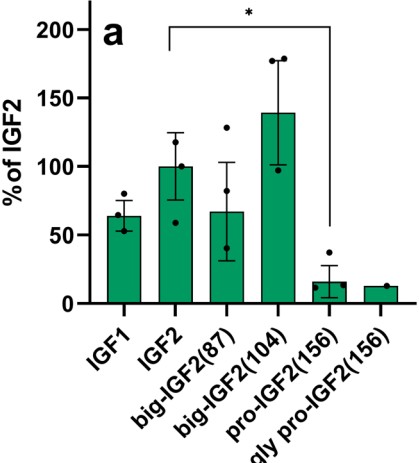
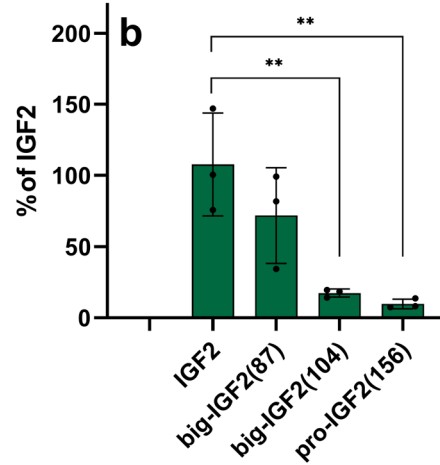

**Fig. 5 Relative binding affinities of IGF2, pro-IGFs and IGF1 to form binary and ternary complexes with IGFBP3 and IGFBP3:ALS. a** Formation of binary complexes with IGBP3; **b** Formation of ternary complexes with IGFBP3:ALS. Data are presented as means ± S.D., relative to the binding affinity of IGF2. Significant differences between marked values determined using Ordinary one-way ANOVA (Dunnett´s multiple comparisons test), *$p < 0.05$, **$p < 0.01$.

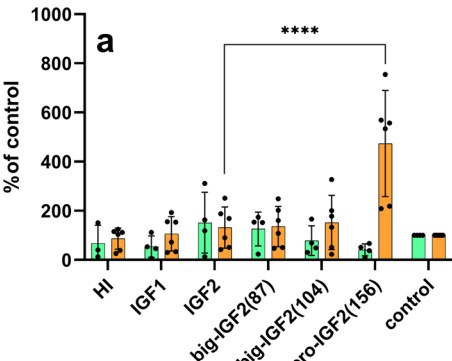
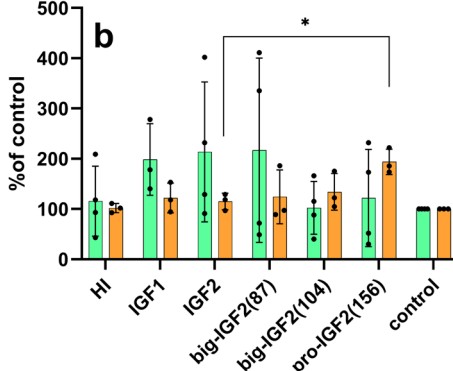

**Fig. 6 Stimulation of phosphorylation of Akt and Erk 1/2.** Stimulation of phosphorylation of intracellular Akt (green) and Erk 1/2 (orange) proteins in murine fibroblasts with deleted *igf1r* gene (R- cells) (**a**) and in MC3T3-E1 preosteoblasts (**b**) by human insulin (HI), IGF1, IGF2, big-IGF2(87), big-IGF2(104), and pro-IGF2(156). Data are presented as means ± S.D., relative to the signal in non-stimulated cells. Significant differences between marked values determined using Ordinary one-way ANOVA (Dunnett´s multiple comparisons test), *$p < 0.05$, **$p < 0.01$, ***$p < 0.001$, ****$p < 0.0001$.

**Pro-IGF2(156) and big-IGF2(104) stimulate intracellular calcium release in R-, MC3T3-E1 and U2OS cells.** It has been indicated that binding of IGF2 to M6P/IGF2R is not only followed by the degradation of this hormone, but that this hormone:receptor coupling also contributes to a variety of physiological events, such as calcium ions' mobilization, or cell migration[35–37]. This would be important, as, for example, changes in intracellular calcium levels are key for activation of ion channels and G protein-coupled receptors. Here, we attempted to probe such hormone:M6P/IGF2R mediated signaling and its impact on intracellular calcium levels in M6P/IGF2R-expressing human osteoblasts U2OS, fibroblast-like cells R- and murine pre-osteoblasts MC3T3-E1 (see above), by testing different proforms of IGF2 in these cell systems. Here, big-IGF2(104) had the same ability to stimulate calcium release as the native IGF2 in all types of cells, while, unexpectedly, the longest pro-IGF2(156) was the most effective stimulator of intracellular calcium release out of all IGF2 forms in both MC3T3-E1 preosteoblasts and U2OS osteoblasts. In contrast, big-IGF2(87) and IGF-1 were completely unable to affect calcium release in all types of cells (Fig. 7).

**Big-IGF2(104) and pro-IGF2(156) significantly increase differentiation of human bone marrow skeletal stem cells (hBMSC-TERT).** Since IGF2 plays a major role in bone growth

and maintenance, the mitogenic and osteogenic activities of IGF2, big-IGF2s and pro-IGF2(156) were investigated. Osteoblast differentiation of human bone marrow skeletal stem cells with overexpression of human telomerase reverse transcriptase gene (hBMSC-TERT)[38] was studied, employing enzymatic assay to measure alkaline phosphatase (ALP) activity (a key enzyme that is important for osteoblast differentiation) (Fig. 8). Furthermore, mineralized matrix formation in hBMSC-TERT differentiated into osteoblasts in vitro by Alizarin Red S staining (Fig. 9) was measured as well. Consistent with the previous experiments and findings described here, pro-IGF2(156) as well as big-IGF2(104) showed higher osteoblast stimulation and differentiation than IGF2 and big-IGF2(87), as quantified by the ALP activity (Fig. 8c and d). The big-IGF2(104) significantly increased ALP activity at concentrations 100 and 500 ng/ml, with pro-IGF2(156) inducing significant ALP increase at 500 ng/ml. In contrast, glycosylated pro-IGF2(156) did not significantly increase ALP activity at either concentration (Fig. 8e). Results with non-glycosylated IGF2 proforms were confirmed by Alizarin Red S staining, where the highest mineralization of matrix was achieved with IGF2 and pro-IGF2(156) at concentrations 500 and 1000 ng/ml, and with big-IGF2(104) at 100 and 500 ng/ml, with big-IGF2(87) remaining inactive at all tested concentrations (Fig. 9). Therefore, these findings suggest a similar effect of the two longer IGF2 proforms on mitogenic and osteoblastic properties in mouse and human cells.

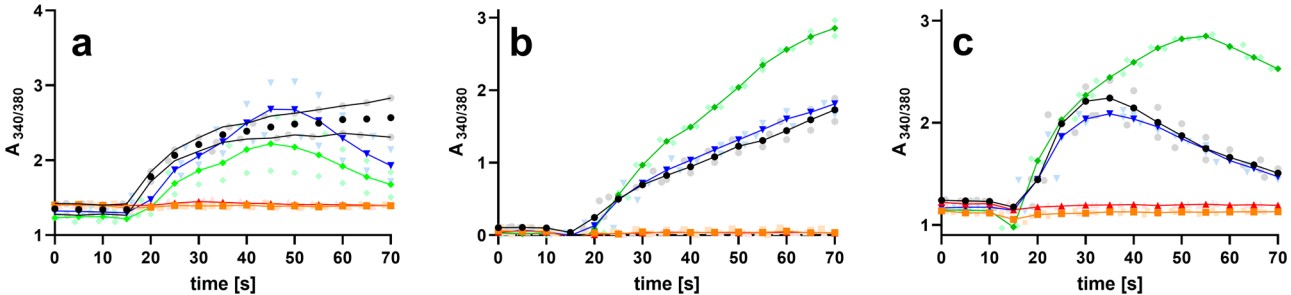

**Fig. 7 Analysis of intracellular calcium mobilization by IGF2, IGF1 and IGF2 prohormones.** R- cells (**a**), U2OS cells (**b**), and MC3T3-E1 cells (**c**) by IGF2 (black), IGF1 (red), big-IGF2(87) (orange), big-IGF2(104) (blue), pro-IGF2(156) (green). Data represent the 340/380 (averages of duplicate measurements plus error ranges) ratio recorded for 70 s by FlexStation 3 and $n = 2$ independent experiments with 2 replicates.

## Discussion

Despite it being more than 30 years since the discovery of several different IGF2-precursor derived forms of this hormone[39], the mechanism of their action and their impact on human physiology are still unknown. It has been suggested that these proforms, which are found in normal blood serum, play an important role in the development and progression of severe diseases, such as non-islet cell tumor hypoglycemia (NICTH) and hepatitis C-associated osteosclerosis (HCAO)[17]. The elevated levels of unprocessed IGF2 forms have been associated with these diseases, but it is still unclear what the exact roles of the individual IGF2 proforms are and what the molecular mechanism of their actions is. These unprocessed forms of pro-IGF2(156) probably result from a defective glycosylation of the IGF2 prohormone, as improper glycosylation prevents the cleavage of a precursor to the mature protein by prohormone convertase, probably pro-protein convertase subtilisin/kexin type 4 (PCSK4)[40].

We prepared and characterized all three known IGF2 proforms, i.e., the full-length protein pro-IGF2(156) and the partially processed big-IGF2(104) and big-IGF2(87). All proteins were in their non-glycosylated forms, as they were suggested to elicit stronger proliferative properties than their glycosylated counterparts[7,10–15,18,20,21,41–44]. In this study, we focused on the detailed characterization of the receptor-binding and activation properties of these non-glycosylated IGF2 proforms towards cognate receptors and specific binding proteins that have not yet been thoroughly characterized. The large number of functional characteristics we determined on all proforms shows these proteins in a broad context of cellular interactions and may indicate which of the proforms is significant in a particular pathological process. For comparison, we added data on binding affinities and proliferation activity of commercially available glycosylated pro-IGF2(156) to explore differences between the glycosylated and non-glycosylated form of this prohormone. Our initial attempts to resolve the 3D structures of IGF2 proforms, using X-ray diffraction or NMR spectroscopy, were not successful. This was probably due to a high flexibility of the individual IGF2 proforms —despite using domain 11 (D11) of the M6P/IGF2R as the stabilizing binding partner[34,45,46]. We therefore measured the CD spectra, to obtain some insight into the representation of the individual secondary structures of the studied proteins. All IGF2 proforms yielded similar CD spectra, similar to that of the mature IGF2, with variation of their intensity. The spectra also indicated that, as the length of the E-domain of IGF2 proforms increases, its helical content decreases, with the increase of the ratio of the unstructured part of the hormone. This is especially prominent for pro-IGF2(156), where the α-helical part makes up only ~15% of the total. The correct folding of individual IGF2 proforms is demonstrated by their potent binding affinities to individual receptors. The activities of misfolded IGF2 analogues have been

previously investigated and it has been shown that such misfolding leads to a substantial decrease in activity[34,47–49]. Spectroscopic results, together with computational predictions, also imply that the peptide chain in the E-domains is completely unstructured and flexible, being detrimental to the structural analyses.

The binding affinities of big-IGF2(87) to IR-A and IR-B were proportional to its ability to activate these receptors, as well to its activation capability for both main signaling pathways analyzed here, PI3K/Akt pathway and Ras/MAPK (Erk 1/2). Moreover, the activity of big-IGF2(87) on these receptors appears to be very similar to the native IGF2, which is consistent with the preliminary results of Marks et al.[6].

Big-IGF2(104) is a significantly more potent activator of the IR-A and IR-B receptors than the native IGF2, which is also reflected in an increased Akt protein phosphorylation by this hormone. This suggests that big-IGF2(104) could influence metabolic and proliferative activities via IR-A and IR-B, although its binding affinity for these receptors remains comparable to that of IGF2. Of all the IGF2 proforms studied here, pro-IGF2(156) has the lowest binding and activation abilities on the IR isoforms, which is reflected in the weaker effects on the Akt protein phosphorylation. Although we determined the binding affinity of the glycosylated form of pro-IGF2(156) for IR-A only and in only one experiment, the results suggest that the binding affinities of the glycosylated and non-glycosylated forms are comparable. Thus glycosylation of pro-IGF(156) would not appear to affect the binding affinity for IR-A. The affinity of unprocessed forms of IGF2 for IGF1R decreases significantly with increasing polypeptide chain length. Here, our results differ from those of Greenall et al.[50], whose glycosylated big-IGF2(87) had the lowest affinity of all glycosylated forms of IGF2 for the purified fragment of the IGF1R ectodomain. In our case, pro-IGF2(156) and gly pro-IGF2(156) have the lowest affinity for IGF1R among all forms tested. This indicates that although chain length plays a role in IGF1R binding, glycosylation of pro-IGF2(156) does not further affect the affinity. The lower binding affinity of pro-IGF2(156) is also reflected in its lower ability to activate IGF1R. Thus, it can be assumed that steric hindrance from the E-domain, which is longer than the matured native IGF2, plays a negative role in binding and activation. The abilities to activate the tyrosine kinase domain of IGF1R for the other two proforms remain similar to IGF2. The lower affinity but similar activation for the shorter big-IGF2(87) and (104) proforms may be due to the fact that neither the D-domain of IGF2 nor the E-domain appear to have direct contacts with the receptor, as indicated by the only available cryo-EM structure of the IGF2:IGF1R complex[51], and some shorter E-domain is tolerated.

There are also no significant differences between native IGF2 and its proforms in the activation of the IGF1R signaling

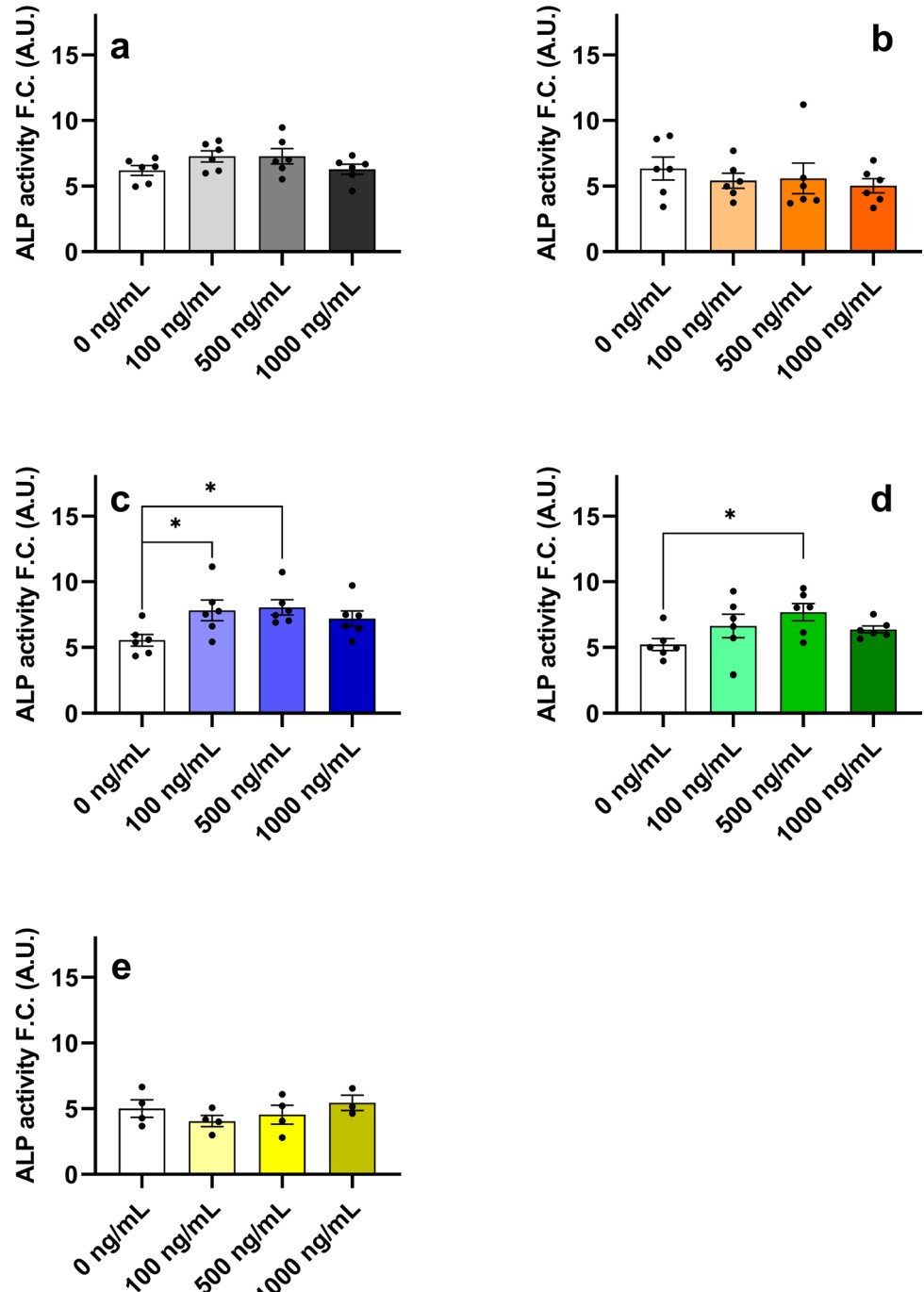

**Fig. 8 Differentiation of hBMSC-TERT osteoblasts after treatment with IGF2 proforms.** IGF2 (**a**), big-IGF2(87) (**b**), big-IGF2(104) (**c**), pro-IGF2(156) (**d**), and gly pro-IGF2(156) (**e**). The data were evaluated using quantification of alkaline phosphatase (ALP) activity, normalized to cell viability presented as a fold change (F.C.) over non-induced cells (day 8), $n = 2$ independent experiments with 6 replicates (only gly pro-IGF2(156) with 4 replicates). Data are presented as mean ± SEM ($n = 6$ (or 4) per group), Ordinary one-way ANOVA (Dunnett´s multiple comparisons test), *$p < 0.05$, **$p < 0.01$, ****$p < 0.0001$. A.U. means arbitrary units.

pathways, except for pro-IGF2(156), which is a significantly weaker inducer of phosphorylation of Erk 1/2[50,51]. Nevertheless, our cumulative data presented here suggest that the effects triggered by the IGF2 proforms through the IGF1R are probably physiologically less significant or important than the consequences of their interaction with IR isoforms.

The characterization of IGF2 proforms with M6P/IGF2R is limited to the work by Greenall et al.[50] However, they studied a glycosylated form of these hormones, while we focused on pathologically more relevant non-glycosylated proforms of IGF2.

We were able to determine the binding affinities of our proteins toward M6P/IGF2R, by using a new in vitro methodology developed in our laboratory[33] and by using cells with a high abundance of M6P/IGF2R, but with negligible levels of IR and IGF1R, which prevents interference and bias of these receptors with M6P/IGF2R binding. Binding of the hormones to both the D11-domain of the M6P/IGF2R and the whole native M6P/IGF2R receptor gave similar results (Fig. 3, Table 1). While the binding affinity of big-IGF2(87) and native IGF2 are similar, the binding affinity of pro-IGF2(156) is lower, with big-IGF2(104)

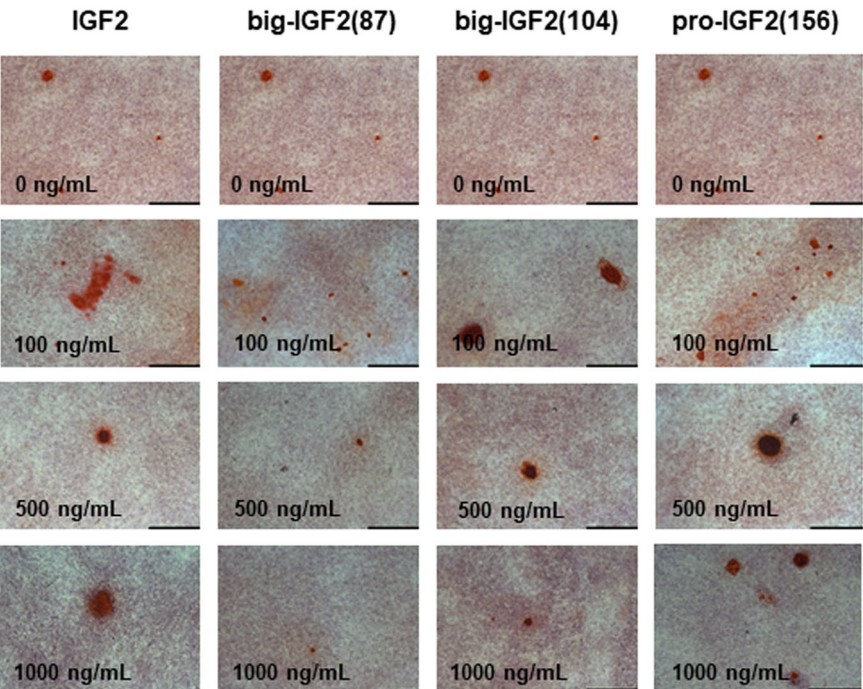

**Fig. 9 Alizarin red S staining of hBMSC-TERT differentiated into osteoblasts.** Alizarin red S staining of hBMSC-TERT differentiated into osteoblasts for 11 days in vitro (scale bar 200 μm); ×10 magnification. The cells were treated with increasing concentrations of non-glycosylated prohormones and native IGF2.

showing an extremely high binding affinity for this receptor (~700–800% of native IGF2). This exceptionally high M6P/IGF2R:big-IGF2(104) binding was observed for both the D11-domain and the entire M6P/IGF2R receptor in cells. Moreover, the big-IGF2(104) and big-IGF2(87) binding to M6P/IGF2R, but not pro-IGF2(156), appeared to follow a two binding sites' mode. This supports the observations in the structural analysis by Wang et al.[52] that IGF2: M6P/IGR2R binding has an allosteric character that involves cooperation of multiple domains of this receptor. The exceptionally high big-IGF2(104):M6P/IGF2R binding affinity is surprising, and its molecular origin is hard to predict, without the actual knowledge of the 3D structure of the complex. To complement the binding abilities of unprocessed IGF2 proforms, we also investigated the binding of glycosylated pro-IGF2(156) to our R -cell system. Surprisingly, we found only a low affinity and the glycosylated ligand was unable to displace $^{125}$I-IGF2 to no more than 50%. Thus, glycosylation seems to be able to affect binding to M6P/IGF2R.

The results of the IGFBP3 binding experiments showed that the ability of IGF2 variants to form binary complexes generally correlates positively with their binding to the M6P/IGF2R, probably due to a similarity of respective binding sites in the hormones. Big-IGF2(87) retained the native IGF2-like ability to form a binary IGFBP3 complex, whereas our pro-IGF2(156), as well as its glycosylated counterpart, showed significantly lower binding affinities for this protein. Interestingly, big-IGF2(104) has an affinity of about 50% higher than IGF2. This agrees with other reports which identified the interaction of the non-glycosylated IGF2 proforms with IGFBP3 but did not specify any $K_d$ value of these interactions[7,53]. One explanation for the apparently higher affinity of big-IGF2(104) for IGFBP3 may be an overlap of its binding sites for M6P/IGF2R and IGFBP3, hence the binding modes to both proteins may be similar[54].

However, the formations of ternary hormone:IGFBP3:ALS complexes by pro-IGF2s show different patterns. Using the immobilized ALS, we demonstrated that the ability of unprocessed forms of IGF2 to form a ternary complex with IGFBP3:ALS decreases with the length of the E- peptide chain, which, again, is supported by previous works, but without the exact $K_d$ values of these interactions[28,53,55,56]. Although we did not determine the ability of glycosylated pro-IGF2(156) to form a ternary complex in our study due to insufficient quantity of material, we can refer to the work of Greenall et al.[50] where authors reported a lower ability of non-glycosylated pro-IGF2(156) to form a ternary complex. In the case of glycosylated, this ability was even much lower.

Normal human serum levels of IGF2 are about 500–1500 times higher than insulin, and only binding of IGF2 to IGFBPs protects the body from permanent hypoglycemia[3]. Therefore, reduced abilities of IGF2 prohormones to form ternary hormone:IGFBP3:ALS complexes can lead to an increase of unbound (or in binary-only complexes) freely circulating pro- and big-IGF2s, which can permeate the capillaries more rapidly, gaining access to membrane receptors more easily than the native IGF2:IGFBP3:ALS complex[18,25]. Thus, this increased tissue-bioavailability of IGF2 prohormones can lead to hypoglycemia, known for example, for the NICTH[17,18,26]. In addition, elevated levels of unbound IGF2 and unprocessed IGF2 proforms have also been shown to cause a marked suppression of secretion of the GH by the pituitary gland, resulting in a marked decrease of GH-dependent proteins such as IGF1, IGFBP3, and ALS[14,20,22]. Furthermore, levels of IGFBP2—not capable of forming a ternary complex—are markedly increased in NICTH, enhancing further IGF2 bioavailability[7,14]. Ultimately, increasing levels of pro- and big-IGF2 suppress PCSK4 expression[40,57], and all together lead to a vicious cycle of high IGF2 and pro-IGF2 levels, triggering severe hypoglycemia.

As IGF2 proforms are suspected of having higher proliferative activity compared to fully matured IGF2, an important experiment was to determine their relative proliferative potential by monitoring the phosphorylation of specific intracellular proteins, intracellular calcium release, ALP activity, and differentiation of

hBMSC-TERT cells. Our results here consistently indicate that pro-IGF2(156), and, to some extent also big-IGF2(104), have the highest growth and osteoblastic potential in both fibroblasts and hBMSC-TERT of murine and human origin. Our observation is in contrast to the study by Khosla et al., who linked the highest proliferative abilities mainly with big-IGF2(87) and big-IGF2(104)[10]. We found, however, that big-IGF2(87) is the least mitogenic molecule, with no activity on calcium release and minimal differentiation activity in the stem cells. Therefore, it seems that big-IGF2(87) probably only appears as a degradation product of pro-IGF2 processing. Although pro-IGF2(156) has lower affinity for IGF1R than big-IGF2(104), this affinity is sufficient to induce its higher proliferative activity, whereas similar proliferative effect of more tightly binding big-IGF2(104) may be reduced by its higher activity for M6P/IGF2R. When comparing properties of the glycosylated and non-glycosylated forms of pro-IGF2(156), the higher ALP activity of non-glycosylated pro-IGF2(156) indicates its higher proliferative ability. On the other hand, the M6P/IGF2R receptor, which normally sequesters IGF2 from the circulation, may partially eliminate the proliferative activity of circulating big-IGF2(104) due to this high binding affinity. This hypothesis is supported by the high abundance of M6P/IGF2R in the cell lines used to examine the proliferative activities of all IGF2 proforms (R-, U2OS, MC3T3-E1 cells). Finally, a higher mitogenicity of pro-IGF2(156) and big-IGF2(104) may result from their lower abilities to form stable binary and ternary complexes, compared to IGF2 and big-IGF2(87).

Our discovery of the abnormally high binding of big-IGF2(104) to the M6P/IGF2R receptor prompts further, rather complex investigations. As the IGF2:M6P/IGF2R interaction is currently considered to be one of the critical factors in consolidating and improving memory, and even synapse growth[58–61], big-IGF2(104) could have therapeutic potential in certain neuropathological cases. However, a potential role of this hormone in tumorigenesis must also be considered.

In summary, our comprehensive study on IGF2 proforms largely expands and clarifies the knowledge about unprocessed, pathologically-relevant non-glycosylated proforms of the IGF2, with the following key findings: (i) the non-glycosylated IGF2 proforms can most probably cause hypoglycemia mainly through binding to IR, whereas their binding affinities and biological effects mediated through IGF1R are overestimated; (ii) the abilities of these hormones to form stable ternary complexes with IGFBP3:ALS markedly decrease with the increase of the length of their E-chains and this can cause their accumulation in the circulation; (iii) pro-IGF2(156) and big-IGF2(104) appear to have the most proliferative and osteoblastic effects of all IGF2 proforms, also including native IGF2; (iv) big-IGF2(104) has a markedly higher binding affinity for M6P/IGF2R of all IGF2 forms; (v) big-IGF2(87) appears to be the least potent of the three proIGF2s, probably only remaining as a degradation product of pro-IGF2 and not a fully biologically active molecule. Therefore, the (pro)IGF2-involving pathogenic scenarios outlined above have clearly been underpinned by our results presented here.

## Methods
### Recombinant production of human big-IGF2s (1-87AA, 1-104AA) and pro-IGF2 (1-156AA). 
All IGF2 proforms were produced using the same technique as we described previously for native IGF2 and its mutants, but with extensive optimization[33,62]. Pro- and big-IGF2s were cloned into a modified pRSFDuet-1 expression vector as the fusion with an N-terminally His6 tagged-GB1 protein and TEV protease cleavage site. The constructs were transformed into *E. coli* BL21 (λDE3) for protein production and fermented until optical density reached $OD_{600} = 1.0$, with further induction by 1 mM IPTG and cultured for an additional 3–4 h. Bacteria were harvested by centrifugation for 20 min at $6000 \times g$ at 4 °C.

The protein constructs are stored in inclusion bodies. To isolate the protein, we resuspended cells in 50 mM Tris-HCl pH 8.7, 50 mM NaCl, 5 mM EDTA, 0.1 mM PMSF and lysed by Emulsiflex1. Cell lysate was centrifuged at $20,000 \times g$ for 20 min. The supernatant was discarded, the pellet dissolved in the same buffer supplemented with 0.1% Triton-X100 and sonicated until dissolved for approximately $3 \times 20$ s. Then the solution was centrifuged at $20,000 \times g$ for 20 min and the same procedure was performed once more, but without Triton-X100. The combined pellet obtained after the last centrifugation was stored at −20 °C. Inclusion bodies were dissolved in reducing and denaturing buffer, composed of 8 M urea, 50 mM Tris-HCl pH 8.0, 300 mM NaCl, 32.5 mM β-ME. The resolubilized protein was incubated for 2 h at 4 °C with slow stirring. The sample was spun down and the desired protein was purified by IMAC, using $Ni^{2+}$-NTA resin on ÄKTA Pure system. The elution fraction containing the desired protein was dialyzed for 16 h at 4 °C with stirring against 50 mM Tris, pH 8.0, 500 mM NaCl, 500 mM *L*-arginine buffer. Longer dialysis caused protein precipitation. Thereafter, the fusion partner was cleaved off with TEV protease and the solution was again purified on $Ni^{2+}$-NTA resin. The collected protein from flow-through was then desalted and purified by RP-HPLC. The molecular weight was confirmed by mass spectrometry. The purity was checked by SDS-PAGE and RP-HPLC. The resulting proteins had more than 95% purity (HPLC).

Glycosylated pro-IGF2(156) was purchased from the company Biomol (No. 97562).

### Circular dichroism and secondary structure analysis. 
IGF2, big-IGF(87) and (104), and pro-IGF2(156) were dissolved in 0.1% $CH_3COOH$ to the final concentration of 0.1 mg/mL and their UV–vis absorption and circular dichroism (CD) spectra were recorded, using the JASCO J-818 spectrometer. All spectra represent an average of 3 consecutive scans acquired in the range of 185–270 nm. All experiments were performed using a 1 mm quartz cell (Hellma Analytics) at room temperature, using the scanning speed of 5 nm/min, and the response time of 16 s. The signal of acetic acid (recorded under the same conditions) was subtracted from the final spectra. The absorption and CD spectra were normalized to concentration (IGF2: $1.34 \times 10^{-5}$ M; big-IGF2(87): $1.03 \times 10^{-5}$ M; big-IGF2(104): $8.36 \times 10^{-6}$ M; pro-IGF2(156): $5.67 \times 10^{-6}$ M), cell length, and the number of amino acids (67, 87, 104, and 156, resp.). The secondary structure content of all proteins was estimated, using the BeStSel[63,64] or K2D3 program[65]. For comparison, we calculated the α-helix content in IGF2 as: $hc = \left(N_{\alpha-helix}/N_{all}\right) \cdot 100$, where $N_{\alpha-helix}$ is the number of residues involved in the α-helical part of the IGF2 structure (2L29.pdb) and $N_{all} = 67$ is the number of all amino acids of IGF2. Note that $N_{\alpha-helix}$ may vary, according to analyzed IGF2 structure (see Table S2). Analogously, we also calculated the hypothetical hc for big-IGF2(87), big-IGF2(104, and pro-IGF2(156), where $N_{\alpha-helix}$ was the same as for IGF2, but $N_{all}$ was 87, 104, and 156, respectively.

3D structures of both big-IGF2s (87 and 104), pro-IGF2(156), and IGF2 as a control were predicted, using ColabFold[66], which combines the homology search of MMseq2[67] with AlphaFold2[68]. Thus, the MMseqs2-based homology search was used to build diverse MSAs; no template mode was applied, and the protein structure prediction was performed using the AlphaFold2-ptm score.

**Cloning and production of D11**. C-terminally His-tagged Domain-11 of human M6P/IGF2R (D11) was prepared according to our previous protocol[33]. Briefly, D11 was cloned into a modified pET24a expression vector. The plasmid was sequenced and verified using Next Generation Sequencing (Eurofins Genomics, Germany). Protein was produced in *E. coli* BL21 (λDE3) and purified. Cultivation was done at 37 °C to reach optical density ~1.0 at 600 nm, followed by an induction with 1 mM IPTG (isopropyl β-D-1-thiogalactopyranoside) and cultured for another 4 h. We then proceeded to the preparation of inclusion bodies.

**Cell cultures**. IM-9 cells (ATCC) were grown in RPMI 1640 medium, supplemented with 10% fetal bovine serum, 2 mM *L*-glutamine, 100 units/mL penicillin, and 100 µg/mL streptomycin in humidified air with 5% $CO_2$ at 37 °C. Mouse embryonic fibroblasts used for binding and signaling were derived from animals with targeted disruption of the IGF-1 receptor gene[69] and stably transfected with expression vectors containing either A (R − /IR-A) or B (R− /IR-B) isoforms of human insulin receptor or human IGF-1 receptor (R+39)[70,71]. As a model for the determination of binding affinities of peptides towards M6P/IGF2R, we used non-transfected R- cells that contain predominantly only this receptor. We also measured saturation binding curves with radiolabeled insulin, IG1 and IGf2 curves to determine the amount of insulin, IGF1 and IGF2 receptors in these cells (see below). The R− /IR-A, R− /IR-B, R+39 and R- cell lines were kindly provided by A. Belfiore (University of Magna Graecia, Catanzaro, Italy) and R. Baserga (Thomas Jefferson University, Philadelphia, PA). Cells were grown in DMEM medium with 5 mM glucose (Biosera), supplemented with 10% fetal bovine serum, 2 mM *L*-glutamine, 0.3 µg/mL puromycin (not added to R- cells), 100 units/mL penicillin, and 100 µg/mL streptomycin in humidified air with 5% $CO_2$ at 37 °C. Mice preosteoblast MC3T3-E1 cells (subclone 4) were purchased from ATCC®. Cells were grown in MEM medium, supplemented with 10% fetal bovine serum, 2 mM *L*-glutamine, 0.3 µg/mL puromycin, 100 units/mL penicillin, and 100 µg/mL streptomycin in humidified air with 5% $CO_2$ at 37 °C. Human osteosarcoma U2OS cells were purchased from ATCC®. Cells were grown in McCoy's 5a medium, supplemented with 10% fetal bovine serum, 2 mM *L*-glutamine, 0.3 µg/mL puromycin, 100 units/mL penicillin, and 100 µg/mL streptomycin in humidified air with 5% $CO_2$ at 37 °C.

A telomerized MSC line (hBMSC-TERT) was used as a model for bone marrow-derived MSCs, as previously described[38]. hBMSC-TERT were maintained in Minimum Essential Medium (MEM), supplemented with 10% fetal bovine serum (FBS) and 1% penicillin/streptomycin (Gibco-Invitrogen, U.S.A.). hBMSC-TERT from passage 41 were used for analyzing their differentiation capacity.

**Determination of binding affinities for IGF1R, IR-A and IR-B**. Binding affinities of ligands for IGF-1R were determined with mouse fibroblasts transfected with human IGF-1R and with deleted mouse IGF-1R (R+39 cells), according to Hexnerova et al.[34] [$^{125}$I]-Iodotyrosyl-IGF1 (2614 Ci/mmol) Perkin Elmer, NEX241025UC) was used as a radiotracer. The dissociation constant of human $^{125}$I-IGF1 for IGF1R was 0.2 nM.

Binding affinities for IR-A were determined with human IR-A in human IM-9 lymphocytes, as described by Morcavallo et al.[72]. Binding affinities for IR-B were determined with mouse fibroblasts transfected with human IR-B and with deleted mouse IGF1R, according to Žáková et al. (Acta D 2014)[73]. For both IR-A and IR-B assays, [$^{125}$I]-monoiodotyrosyl-TyrA14-insulin (2200 Ci/mmol), prepared as described by Asai et al.[74], was used

as a radiotracer. The dissociation constant of human $^{125}$I-insulin for IR-A and IR-B was 0.3 nM.

**Determination of binding affinities for D11 of M6P/IGF2R (D11:M6P/IGF2R)**. The binding affinities of IGF2 and pro-IGF2s for immobilized D11 were determined, as described by Potalitsyn et al.[33] [$^{125}$I]-monoiodotyrosyl-Tyr2-IGF2 (2200 Ci/mmol)[33] was used as a radiotracer and its $K_d$ for immobilized D11 was 2 nM.

**Determination of binding affinities for M6P/IGF2R in R- cells**. Firstly, we determined if R- cells (see above) represent a suitable model for measuring the binding affinities of IGF2 and its derivatives. We performed saturation binding experiments with radiolabeled IGF2, insulin, or IGF1. For this, 14,000 R- cells were seeded in each well in complete medium 24 h before the experiment. The cells were incubated in a total volume of 250 µl of a binding buffer (100 mM HEPES pH 7.6, 100 mM NaCl, 5 mM KCl, 1 mM EDTA, 1.3 mM $MgSO_4$, 10 mM glucose, 15 mM sodium acetate and 1% bovine serum albumin) with various concentrations (0–5 nM) of [$^{125}$I]-monoiodotyrosyl-Tyr2-IGF2 (2200 Ci/mmol), [$^{125}$I]-mono-iodotyrosyl-TyrA14-insulin (2200 Ci/mmol) or [$^{125}$I]- iodotyrosyl-IGF-1 (2614 Ci/mmol) for 16 h at 5 °C. Thereafter, the wells were washed twice with saline. The bound proteins in the wells were solubilized twice with 500 µl of 0.1 M NaOH that was collected. Bound radioactivity was determined in the γ-counter. Nonspecific binding was determined by measuring the remaining bound radiotracer in the presence of 10 µM unlabeled IGF2, insulin or IGF-1 for each tracer concentration. Each experiment was performed 3 times and the results were evaluated in GraphPrism 8, using non-linear regression considering binding to one site.

Next, the determination of the binding affinities of ligands for M6P/IGF2R was performed by the competition of increasing concentrations of a cold ligand ($10^{-12}$–$10^{-6}$ M) with a constant concentration (43,000 cpm) of [$^{125}$I]-monoiodotyrosyl-Tyr2-IGF2 (2200 Ci/mmol) for M6P/IGF2R in R- cells, using the procedure described above. The dissociation constant of $^{125}$I-IGF2 for M6P/IGF2R was 8.2 nM.

**Determination of binding affinities of ligands for formation of binary complex with IGFBP3**. Firstly, we performed saturation binding experiments for the binding of radiolabeled IGF2 to immobilized IGFBP3 and determined the $K_d$ value for this interaction. His-tagged IGFBP3 (25 ng/well, 7.5 nM, Sino Biological) was immobilized on a 96-well plate via neutravidin and the nickel-charged-NTA-decorated iBodies® (IOCB Tech, Prague, Czech Republic, https://www.uochb.cz/en/iocb-tech) containing covalently bound biotin, according to the procedure described for immobilization of D11 by Potalitsyn et al.[33] Immobilized IGFBP3 was incubated in a total volume of 100 µl of the binding buffer (100 mM HEPES pH 7.6, 100 mM NaCl, 5 mM KCl, 1.3 mM $MgSO_4$, 10 mM glucose, 15 mM sodium acetate and 1% bovine serum albumin) with various concentrations (0–5 nM) of [$^{125}$I]-monoiodotyrosyl-IGF2 (2200 Ci/mmol) for 16 h at 5 °C. Thereafter, the wells were washed twice with the TBS buffer without Tween 20. The bound proteins in the wells were solubilized twice with 300 µl of 0.1 M NaOH that was collected. Bound radioactivity was determined in the γ-counter. Nonspecific binding was determined by measuring the remaining bound radiotracer in the presence of 10 µM unlabeled IGF2 for each tracer concentration. The experiment was performed 3 times.

Next, determination of the binding affinities of ligands for the formation of a binary complex was performed by the competition of increasing concentrations of a cold ligand ($10^{-12}$–$10^{-6}$ M) with a constant concentration (43,000 cpm) of [$^{125}$I]-monoiodo-tyrosyl-Tyr2-IGF2 (2200 Ci/mmol) for immobilized IGFBP3,

using the procedure described above. The dissociation constant of [125]I-IGF2 for IGFBP3 was 0.56 nM.

**Determination of binding affinities of ligands for formation of ternary complex with IGFBP3 and ALS.** The affinity of hormones for the formation of a ternary complex with IGFBP3 and ALS was measured using immobilized His-tagged ALS protein. Firstly, we determined the binding affinity of radiolabeled [[125]I]-monoiodotyrosyl-Tyr2-IGF2 for the ternary complex in a saturation binding experiment. Neutravidin was adsorbed on to the bottom of a 96-well plate in 50 mM borate buffer pH 9.5. The wells were then blocked with IgG-free casein for 3 h. His-tagged ALS protein (R&D Systems, 10 ng/well, 1.5 μM) was immobilized in wells via neutravidin and the nickel-charged-NTA-decorated iBodies® (IOCB Tech, Prague, Czech Republic, https://www.uochb.cz/en/iocb-tech) containing covalently bound biotin, according to the procedure described for immobilization of D11 by Potalitsyn et al.[33] After immobilization of ALS, the wells were incubated with guanine nucleotide-binding protein-1 with N-terminal His$_6$-tag for 1 h to block all potentially remaining free binding sites. Then, IGFBP3 (without His-tag, 25 ng/well, 7.5 nM, Sino Biological) was added in a total volume of 100 μl of the binding buffer (100 mM HEPES pH 7.6, 100 mM NaCl, 5 mM KCl, 1.3 mM MgSO$_4$, 10 mM glucose, 15 mM sodium acetate and 1% bovine serum albumin) to immobilized ALS to preform a binary IGFBP3:ALS complex. Then, various concentrations (0-5 nM) of [[125]I]-monoiodotyrosyl-IGF2 (2200 Ci/mmol) were added in a minimum volume of the binding buffer and the mixture was incubated for 16 h at 5 °C. Thereafter, the wells were washed twice with the TBS buffer without Tween 20. The bound proteins in the wells were solubilized twice with 300 μl of 0.1 M NaOH that was collected. Bound radioactivity was determined in the γ-counter. Nonspecific binding was determined by measuring the remaining bound radiotracer in the presence of 10 μM unlabeled IGF2 for each tracer concentration. The experiment was performed three times.

Next, determination of the binding affinities of ligands for the formation of a ternary complex was performed by the competition of increasing concentrations of a cold ligand ($10^{-12}$–$10^{-6}$ M) with a constant concentration (43,000 cpm) of [[125]I]-monoiodotyrosyl-Tyr2-IGF2 (2200 Ci/mmol) for the immobilized IGFBP3:ALS complex as described above. The dissociation constant of [125]I-IGF2 for IGFBP3 was 5.9 nM.

**Analysis of binding data and statistical evaluation of binding affinities.** The binding data obtained from all binding assays were analyzed. The dissociation constant ($K_d$) values were determined with GraphPad Prism 8, using a nonlinear regression method, a one-site fitting program and taking into account the potential depletion of free ligand. The individual binding curves of each peptide for each receptor were determined in duplicate points, and the final dissociation constant ($K_d$) was calculated from at least three ($n \geq 3$) binding curves (each curve giving a single $K_d$ value), determined independently and compared to binding curves for human insulin or human IGF-1, depending on the type of receptor. Relative binding affinities were calculated as ($K_d$ of the native hormone/$K_d$ of analog) × 100 (%). The relative binding affinity S.D. values of analogs were calculated as S.D. = $K_d$ of the native hormone/$K_d$ of peptide ×100 × $\sqrt{[(\text{S.D. native}/K_d \text{ native})^2 + (\text{S.D. analog}/K_d \text{ analog})^2]}$.

**Receptor phosphorylation assay and signaling pathways' activation.** Cell stimulation and detection of receptor phosphorylation were performed as described previously[62], using mouse fibroblasts (IR-A, IR-B and R+39). Western blots were similarly

used to study signaling pathways activated by insulin, IGF1, IGF2 and IGF2 proforms in the R- and MC3T3-E1 cells. All types of cells were stimulated with 10 nM concentrations of the ligands for 10 min. Proteins were routinely analyzed using immunoblotting. The membranes were probed with anti-phospho-IGF-1Rβ (Tyr1135/1136)/IRβ (Tyr1150/1151) (19H7) Rabbit mAb (Cell Signaling Technology #3024, dilution 1:1000), anti-phospho-Akt (Thr308) (C31E5E) Rabbit mAb (Cell Signaling Technology #2965, dilution 1:1000), anti-phospho-p44/42 MAPK (Erk 1/2) (Thr202/Tyr204) (E10) Mouse mAb (Cell Signaling Technology #9106, dilution 1:1000). The secondary antibodies used were anti-Rabbit IgG (whole molecule), Peroxidase conjugate developed in goat (Sigma A-9169, dilution 1:80,000) and anti-Mouse IgG (whole molecule), Peroxidase antibody produced in rabbit (Sigma A-9044, dilution 1:20,000). Each experiment was repeated four times. The data were expressed as the contribution of phosphorylation relative to the human insulin (R-/IR-A, R-/IR-B) or IGF1 (IGF1R) signal in the same experiment or relative to non-stimulated cells (R- and E1). Mean ± S.D. ($n = 4$) values were calculated. The significance of the changes in stimulation of phosphorylation in relation to insulin was calculated, using the t-test comparing all analogs versus control i.e. insulin, IGF1 or non-stimulated cells.

The quantity of IR, IGF1R and IGF2R in employed cell lines (R-, IR-A, IR-B, R+39, MC3T3-E1 and U2OS) was compared, using antibodies detecting whole receptors, namely anti-Insulin Receptor β (4B8) Rabbit mAb (Cell Signalling Technology #3025, dilution 1:1000), anti-IGF1R β (111A9) Rabbit mAb (Cell Signaling Technology #3018, dilution 1:1000) and anti-M6P/IGF2R (D3V8C) Rabbit mAb (Cell Signaling Technology #14364, dilution 1:1000). Anti-actin (20–33) antibody developed in Rabbit, Sigma A-5060, dilution 1:500. Anti-actin (20–33) antibody (Sigma, #A5060, dilution 1:500) was used as a loading control.

The densities of individual bands on blots were quantified by Image Lab Software, version 5.2.1. (Bio-Rad), which is a ChemiDoc integrated software. The densities were related to the signal of insulin (100%) (IR-A and IR-B cells) or IGF-1 (100%) (R+39 cells) in every gel.

**Calcium release.** Calcium measurements in human U2OS cells (ATCC) were done in McCoy Medium (Termo-Fisher Scientific), supplemented with 10% fetal bovine serum, 1% penicillin/streptomycin, and 1% glutamine maintained at 37 °C and 5% CO$_2$. Intracellular calcium levels were determined, using the Fura-2 QBT™ Calcium Kit (Molecular Devices). Cells were seeded (40,000 cells/well in 200 μl media) on black, glass-bottom, 96-well plates (Cellvis). Experiments were performed 24 h after seeding. Cells were loaded for 1 h at 37 °C with Fura-2 QTB HBSS in Hepes buffer, supplemented with 2.5 mM probenecid. Intracellular Ca$^{2+}$ was assessed by measuring changes in fluorescence with a FlexStation 3 reader (Molecular Devices) at RT. Measurements were recorded at 340/510 nm and 380/510 nm (excitation/emission) every 5 s for a total of 110 s. Effects of hormones were measured at $10^{-3}$–$10^{-6}$ M concentrations.

**Osteoblast (OB) differentiation.** hBMSC-TERT were plated at a density of 20,000 cells/cm$^2$ (Alizarin Red S staining) and 5000 cells/well (ALP activity assay) in MEM medium (Gibco), supplemented with 10% FBS (Gibco) and 1% P/S (Gibco). One day after seeding, the medium was replaced with OB induction medium, composed of basal medium, supplemented with 10 mM B-glycerophosphate (Sigma-Aldrich), 10 nM dexamethasone (Sigma-Aldrich), 50 μg/mL vitamin C (Sigma-Aldrich), 10 nM vitamin D3 (Sigma-Aldrich) and treated with 100 ng/ml, 500 ng/

ml and 1 µg/ml IGF2, big-IGF2(87), big-IGF2(104), pro-IGF2(156), or vehicle (acetic acid) as a control. The medium was changed every other day for 8 days (ALP activity assay) or 11 days (Alizarin Red S staining).

**Alizarin Red S staining**. Mineralized matrix formation at Day 11 of OB differentiation was measured, using Alizarin Red S staining[75]. Cells were fixed with 70% ice-cold ethanol for 1 h at −20 °C, before addition of Alizarin Red S Solution (AR-S) (Sigma-Aldrich). The cells were stained for 10 min at room temperature (RT). Excess dye was washed with distilled water and with phosphate buffered saline (PBS) (Gibco) to reduce non-specific AR-S stain.

**Alkaline phosphatase (ALP) activity assay**. ALP activity and cell viability assay were quantified at Day 8 of OB differentiation, in order to normalize the ALP activity data to the number of viable cells. Cell viability assay was measured, using Cell Titer-Blue Assay Reagent (Promega) at fluorescence intensity (579Ex/584Em). ALP activity was detected by absorbance at 405 nm, using p-nitrophenyl phosphate (Sigma-Aldrich) as substrate[75].

## Data availability

Source data used for preparation of this paper are provided with this paper and in the Supplementary electronic information. All source data underlying the graphs shown in the main figures are uploaded as supplementary data.

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

## Acknowledgements

This work was supported by the Czech Science Foundation grant No. 19-14069S (to L.Z.), by the European Regional Development Fund, OP RDE, Project: "Chemical Biology for drugging undruggable targets (ChemBioDrug)" (No. CZ.02.1.01/0.0/0.0/16_019/0000729), by the project National Institute for Research of Metabolic and Cardiovascular Diseases (Program EXCELES, ID Project No. LX22NPO5104, Funded by the European Union-Next Generation EU), and by the Academy of Sciences of the Czech Republic (Research Project RVO:6138963, support to the Institute of Organic Chemistry and Biochemistry). A.M.B. was supported by MRC grant MR/R009066/1 and BBSRC grant BB/W003783/1.

## Author contributions

P.P., I.S., M.T., J.K., J.J. and L.Z. designed the research; P.P., L.M., I.S., M.T., M.F., M.C., A.M., J.K., T.T. and L.Z. performed the research; P.P., L.M., I.S., M.T., M.F., M.C., A.M., J.K., J.J. and L.Z. analyzed the data, P.P., I.S., M.T., A.M.B., J.K., J.J. and L.Z. wrote the manuscript.

## Competing interests

The authors declare no competing interests.
