## [Peer Review File · Communications Biology]

Reviewers' comments:

Reviewer #1 (Remarks to the Author):

This manuscript represents a significant body of work to analyse the biological activity of non-glycosylated IGF2 prohormones. An understanding of their mechanism of action will have relevance to understanding their roles in diseases such as non-islet-cell tumor hypoglycemia and hepatitis C-associated osteosclerosis in which they are highly expressed. A series of assays to measure binding to receptors and IGF2R was undertaken as well as an assessment of biological activity through receptor, Akt and Erk1/2 activation as well as calcium mobilisation and cellular differentiation. The manuscript provides some interesting findings. There is some enthusiasm shown by the authors leading to some over statement of the significance of the results which I would recommend revising, and unfortunately there is no raw data provided for any of the signalling immunoblots.

Comments/suggestions to be addressed:

The CD spectrum of Pro-IGF2(156) is very different to that of IGF2 and most of the other proteins tested. There is not only a shift in intensity at 222nm where helical content is measured but also a leftward shift of the spectrum which is reminiscent of the disulfide swap form of IGF-I. Are you confident that this peptide is correctly folded? Is the MS data accurate enough to be sure you have all disulfides formed and how would you know if this is actually not a disulfide swap form?

Raw data ie all uncropped blots or at minimum representative blots for receptor activation, Akt and Erk signalling need to be included in the supplementary file.

What secondary antibodies were used for the immunoblots? It would be best to provide a table of antibodies used and what concentrations were used. What was the method of quantitation of blots?

Title line 124 "Big-IGF2(104) binds and activates IR-A more efficiently than mature IGF2." is misleading as the affinity of Big-IGF2(104) is the same as the affinity of IGF2 for the IR-A. It is only the activation potency that differs.

Typo Figure 3 legend line 144 "The data on Erk 1/2 are not shown, as they were similar to data on Erk 1/2."

Do the R- fibroblasts express endogenous mouse insulin receptor and could this account for the biphasic nature of the competition curves in Figure S6D?

In Figure 6, assuming the control represents unstimulated cells then the only data points that are significantly above basal for Akt are pro IGF2 156, and for Erk none seem to be above control. Is this correct? Again, there are no blots provided as evidence of this data and they should be included. Perhaps reword the title "Pro-IGF2(156) can stimulate phosphorylation of Erk1/2 in R-cells and MC3T3-E1 preosteoblast more efficiently than other IGF-2 proforms." as none of the other forms stimulate Erk1/2.

Reviewer #2 (Remarks to the Author):

The authors of the article titled "Non-glycosylated IGF2 prohormones are more mitogenic than native IGF2 Pavlo Potalitsyn" produced three non-glycosylated forms of IGF2-related proteins and characterized them using a battery of experiments including HPLC, Mass Spec, CD, cell-based binding, and functional assays. The authors have concluded/speculated that the non-glycosylated IGF2 analogs

can most probably cause hypoglycemia mainly through binding to the insulin receptors; the abilities of the IGF2 variants to form ternary 450 complexes with IGFBP3:ALS markedly decrease with the increase of the size of the proteins; and pro-IGF2(156) and Big-IGF2(104) appear to have the most proliferative effects of all IGF2 variants. I want this paper to be published after major revision.

1) One major limitation of this study is the lack of glycosylated IGF2 variants that include Big-IGF2 (87, 104, and 156) and their comparative studies with non-glycosylated variants they produced in this study. Can authors produce and study them and include the data? Alternatively, can they buy these glycosylated variants and study them?

2) The CD experiments only measure the secondary structures of the proteins. How do authors confirm the correct disulfide folding of the synthetic IGF-2 proteins? Could they have done tryptic digestion mapping?

3) Please include the solvent gradient of the HPLC run in Figure S1 and explain why the retention time of Big-IGF2 proteins is not increasing (or decreasing) with the size if they have used the same HPLC column and the same gradient (Figure S1).

4) It seems that the observed mass data do not exactly match the calculated mass data. Please comment on this including the calculated mass of the proteins in the legend of Figure S2.

5) Please elaborate on the abbreviation of ESI (supporting info, Line 115, page 6) as it contradicts ESI (ESI-MS line 108, page 6).

6) On page 7, line 124: The headline "Big-IGF2(104) binds and activates IR-A more efficiently than mature IGF2" contradicts the following sentence "Big-IGF2(104) and big-IGF2(87) bind both isoforms of the insulin receptor (IR-A, IR-B) similarly to native IGF2" ... please revise this.

7) The authors mixed up the names throughout the texts. I recommend that they always use IGF-2, Pro-IGF2(156), Big-IGF(104) and Big-IGF2(87). See the names in Table 1, for example.

Reviewer #3 (Remarks to the Author):

This interesting, well-designed and comprehensive study investigates the role, binding properties and signal transduction of the IGF2 precursor, pro-IGF2 (156), and two IGF2 proforms, big-IGF2 (87) and big-IGF2 (104) that result from incorrect processing and glycosylation of the precursor, and are found at abnormally high levels in certain diseases. The results clarify and improve our current knowledge of the biological importance and pathogenicity of the individual IGF2-derived proforms of this hormone. First, I would like to address the background information given about the M6P/IGF2 receptor and the postnatal role of IGF2, which could be improved.

The introduction starts with two sweeping statements that are not referenced:

1. "Insulin-like growth factor-2 (IGF-2) is involved in the growth and function of almost every organ in the human body, also having a therapeutic potential in a variety of clinical disorders." This statement needs references. The postnatal role of IGF2 is far from clear and has been much less explored than that of IGF1. While IGF2 remains highly expressed in humans, increasing until puberty and showing only a slight decrease with aging, in rodents the *Igf2* gene is downregulated after birth.

2. "Most of IGF2's biological actions are mediated by its binding and signaling through (...) a structurally distinct mannose-6-phosphate/IGF2 receptor". This statement also needs references. It is clear that evolutionarily, the cation-independent M6P receptor in vertebrates is not a signaling receptor, but a transmembrane glycoprotein that targets newly synthesized lysosomal hydrolases to the lysosomes. It is only in mammals that this receptor acquires a module that binds IGF2, targeting IGF2 to the lysosomes where it is degraded, thus controlling IGF2 levels to prevent organ overgrowth.

The authors may want to consider looking at and citing some classical basic references that address these issues:

Baker J, Liu JP, Robertson EJ, Efstratiadis A (1993). Role of insulin-like growth factors in embryonic and postnatal growth. *Cell* 75:73-82.

Morrione A, Valentinis B, Xu SQ, Yumet G, Louvi A, Efstratiadis A, Baserga R (1997). Insulin-like growth factor II stimulates cell proliferation through the insulin receptor. *Proc Natl Acad Sci USA* 94:

3777-3782.

Ludwig T, Eggenschwiller J, Fisher P, D'Ercole AJ, Davenport ML, Efstratiadis A (1996). Mouse mutants lacking the type 2 IGF receptor (IGF2R) are rescued for perinatal lethality in *Igf2* and *Igf1r* null backgrounds. *Dev Biol* 177:517-535.

Louvi A, Accili D, Efstratiadis A (1997). Growth promoting interaction of IGF-II with the insulin receptor during mouse embryonic development. *Dev Biol* 189: 33-48.

Wolf E, Hoeflich A, Lahm H (1998). What is the function of IGF-II in postnatal life? Answers from transgenic mouse models. *Growth Hormone and IGF Research* 8:185-193.

Otherwise, I find the study logically constructed and well written, and well illustrated. The wide range of techniques used are well described. The analyses are very quantitative. The statistical analyses are adequate.

The key findings are as follow, underpinning the previously proposed (pro)IGF2-involving pathogenic scenarios:

- (i) The non-glycosylated IGF2 proforms probably cause hypoglycemia through binding to the insulin receptor;
- (ii) The abilities of these hormone to form ternary complexes with IGFBP3:ALS markedly decrease with the increase in the length of their E-chains;
- (iii) Pro-IGF2(156) and big-IGF2 (104) have the most proliferative and osteoblastic effects of all IGF2 proforms, also including native IGF2;
- (iv) Big-IGF2(104) has a markedly higher binding affinity for M6P/IGF2R of all IGF2 forms;
- (v) Big-IGF2(87) appears to be the least potent of the three proIGF2s, probably only remaining as a degradation product of pro-IGF2 and not a fully biologically active molecule.

These represent clear advances in knowledge.

A few minor comments:

Line 144 in legend of Figure 3: "The data on Erk 1/2 are not shown, as they are very similar to data on Erk 1/2" is nonsensical.

Lines 707-709 in Authors contributions: L. should be L.Z.

Line 764 in References: apf should be Zapf.

Reviewer: Pierre De Meyts

We would like to thank you and the reviewers for the careful assessment of our work and detailed comments. These comments were very valuable, helping us to clarify the major points of the paper and to remove ambiguities. We hope that we have addressed all concerns raised by the reviewers and implemented their suggestions effectively. Here are the point-by-point answers to the reviewers' questions.

Reviewers' comments:

Reviewer #1:

1. The CD spectrum of Pro-IGF2(156) is very different to that of IGF2 and most of the other proteins tested. There is not only a shift in intensity at 222nm where helical content is measured but also a leftward shift of the spectrum which is reminiscent of the disulfide swap form of IGF-I. Are you confident that this peptide is correctly folded? Is the MS data accurate enough to be sure you have all disulfides formed and how would you know if this is actually not a disulfide swap form?

We have discussed misfolded IGF2 analogues in our previous paper where we precisely distinguished a swap from a correctly folded IGF2 analogue (Hexnerová, J. Biol. Chem. 2016). A crucial feature of misfolded (swapped) IGFs is the almost complete loss of binding affinity, as we and others have previously shown (Gill, Protein Eng. 1999; Sohma, Angew. Chem. Int. Ed. 2008; Hexnerová, J. Biol. Chem. 2016). Our prepared IGF2 proforms (this work) always have sufficient affinity relative to native IGF2. The observed change in the CD spectrum of pro-IGF2(156) compared to the others is mainly due to the presence of the unstructured E domain of pro-IGF2(156), which is longer (89 amino acids) than the entire native IGF2 (67 amino acids). This leads to a change in abundance of individual secondary structures, an overall decrease of the α -helical content, and affects the spectrum. Changes in CD spectra of big-IGFs are not as significant because their prolongation compared to IGF2 is lower than that of pro-IGF2(156). The decrease in estimated helix abundance corresponds well to the increase in the length of the IGF2 proforms. Moreover, in the paper by Gill et al (Gill, Protein Eng. 1999), authors recorded CD spectra of correctly and misfolded IGF1 and found that misfolded IGF1 exhibited a lower α -helical content of about 5% compared to correctly folded IGF1 (we observed a decrease of ~15%). In that case, considering the aforementioned change in the overall α -helix abundance due to the unstructured E-domain in pro-IGF2(156), the α -helical content of possibly misfolded pro-IGF2(156) should be even lower than we observed.

The discussion of the manuscript was edited, and relevant citations were added.

2. Raw data i.e., all uncropped blots or at minimum representative blots for receptor activation, Akt and Erk signalling need to be included in the supplementary file.

Representative blots were added (Figure S12) to the supplemental material.

3. What secondary antibodies were used for the immunoblots? It would be best to provide a table of antibodies used and what concentrations were used. What was the method of quantitation of blots?

We have added all the information about the primary and secondary antibodies used in this study to the Method chapter, and we have also added information about the Software that was used to evaluate the blots.

4. Title line 124 "Big-IGF2(104) binds and activates IR-A more efficiently than mature IGF2." is misleading as the affinity of Big-IGF2(104) is the same as the affinity of IGF2 for the IR-A. It is only the activation potency that differs.

Yes, the reviewer is right, it has been corrected in the text.

5. Typo Figure 3 legend line 144 "The data on Erk 1/2 are not shown, as they were similar to data on Erk 1/2."

It has been corrected.

6. Do the R- fibroblasts express endogenous mouse insulin receptor and could this account for the biphasic nature of the competition curves in Figure S6D?

Based on the saturation curves we performed on R cells (Figure S7), the presence of the insulin receptor is so low that it is unlikely that the biphasic curve is due to the presence of the insulin receptor.

7. In Figure 6, assuming the control represents unstimulated cells then the only data points that are significantly above basal for Akt are pro IGF2 156, and for Erk none seem to be above control. Is this correct? Again, there are no blots provided as evidence of this data and they should be included. Perhaps reword the title "Pro-IGF2(156) can stimulate phosphorylation of Erk1/2 in R-cells and MC3T3-E1 preosteoblast more efficiently than other IGF-2 proforms." as none of the other forms stimulate Erk1/2.

In Figure 6 there is indeed significant phosphorylation of Erk1/2 (not Akt) only by pro-IGF2(156), and by none of our other IGF2 proforms. The title and description in the original text should be correct.

Reviewer #2:

1. One major limitation of this study is the lack of glycosylated IGF2 variants that include Big-IGF2 (87, 104, and 156) and their comparative studies with non-glycosylated variants they produced in this study. Can authors produce and study them and include the data? Alternatively, can they buy these glycosylated variants and study them?

We agree with the referee that the comparison with glycosylated proforms would be beneficial. However, the preparation of such glycosylated IGF2 proforms is very complex and would be the work subject of an whole entirely new study. To prepare glycosylated proteins, we would have to choose a different expression system and also completely different conditions than the proteins we have been preparing. Moreover, this would mean repeating the excellent work of Greenall et al (JBC, 2013), which was not our aim. Nevertheless, we tried to accommodate Reviewer's request and purchased the only available glycosylated proform of IGF2, the whole pro-IG2(156). With a limited amount of this protein we performed some biological essays that we found interesting to compare with its non-glycosylated form. These results are included in the manuscript.

2. The CD experiments only measure the secondary structures of the proteins. How do authors confirm the correct disulfide folding of the synthetic IGF-2 proteins? Could they have done tryptic digestion mapping?

The answer is given as in Reviewer #1

3. Please include the solvent gradient of the HPLC run in Figure S1 and explain why the retention time of Big-IGF2 proteins is not increasing (or decreasing) with the size if they have used the same HPLC column and the same gradient (Figure S1).

Gradient information is added to Figure S1. In general, the retention time is affected by the hydrophobicity of individual IGF2 profiles and not by the size of the molecule, although longer peptides usually have higher retention on RP-columns. In this case, we assume that the effect of lengthening the E domain is compensated by its distinctly hydrophilic nature.

4. It seems that the observed mass data do not exactly match the calculated mass data. Please comment on this including the calculated mass of the proteins in the legend of Figure S2.

The mass spectrum of big-IGF2(87) analyzed by ESI (electrospray ionization) is very accurate and the mass obtained matches exactly its theoretical monoisotopic mass (in Figure S2A, the operator "deconvoluted" the spectrum and subtracted the hydrogen atom). For large molecules the ESI method is not applicable. For spectra obtained by the MALDI method, in our experience and experience of specialist in our MS department, the molecular mass can vary by $\pm 8-10$ Da for larger molecules. For big-IGF2(104) there is a very good agreement between theory and experiment (in Figure S2B). In Figure S2C, the difference between the theoretical (17635) and measured data (17642) for pro-IGF2(156) the difference is 7 Da, which is in a reasonable range. Higher molecular species (e.g. 17739 can represent molecule with few Na⁺ or K⁺ atoms bound).

5. On page 7, line 124: The headline "Big-IGF2(104) binds and activates IR-A more efficiently than mature IGF2" contradicts the following sentence "Big-IGF2(104) and big-IGF2(87) bind both isoforms of the insulin receptor (IR-A, IR-B) similarly to native IGF2" ... please revise this.

Corrected

6. The authors mixed up the names throughout the texts. I recommend that they always use IGF-2, Pro-IGF2(156), Big-IGF(104) and Big-IGF2(87). See the names in Table 1, for example

Corrected

Reviewer #3:

1. "Insulin-like growth factor-2 (IGF-2) is involved in the growth and function of almost every organ in the human body, also having a therapeutic potential in a variety of clinical disorders." This statement needs references. The postnatal role of IGF2 is far from clear and has been much less

explored than that of IGF1. While IGF2 remains highly expressed in humans, increasing until puberty and showing only a slight decrease with aging, in rodents the Igf2 gene is downregulated after birth.

We have edited this paragraph and added respective references.

2. "Most of IGF2's biological actions are mediated by its binding and signaling through (...) a structurally distinct mannose-6-phosphate/IGF2 receptor". This statement also needs references. It is clear that evolutionarily, the cation-independent M6P receptor in vertebrates is not a signaling receptor, but a transmembrane glycoprotein that targets newly synthesized lysosomal hydrolases to the lysosomes. It is only in mammals that this receptor acquires a module that binds IGF2, targeting IGF2 to the lysosomes where it is degraded, thus controlling IGF2 levels to prevent organ overgrowth.

Recently, the preview of the M6P/IGF2R signaling properties has changed (El-Shewy H., J. Biol. Chem., 2007, Alberini C. Trends Neurosci., 2023), but we have edited the paragraph and added citations.

3. The authors may want to consider looking at and citing some classical basic references that address these issues:

Thank you for adding useful citations, we have added most of them to the manuscript.

A few minor comments:

Line 144 in legend of Figure 3: "The data on Erk 1/2 are not shown, as they are very similar to data on Erk 1/2" is nonsensical.

Corrected

Lines 707-709 in Authors contributions: L. should be L.Z.

Corrected

Line 764 in References: apf should be Zapf.

Corrected

REVIEWERS' COMMENTS:

Reviewer #1 (Remarks to the Author):

The authors have addressed the comments adequately.

Reviewer #2 (Remarks to the Author):

The authors addressed my comments and I am happy for this paper to be accepted.